# Admission screening for extended-spectrum cephalosporin-resistant and carbapenem-resistant Enterobacterales colonization at a referral hospital in Botswana: A one-year period-prevalence survey, 2022–2023

Tshiamo Zankere [1]*[◉], Kwana Lechiile [1,2][◉], Kitso Mokgwathi[1], Nametso Tlhako[1], Banno Moorad[1], Tlhalefo D. Ntereke[1], Teresia Gatonye[1], Ebbing Lautenbach[3,4], Melissa Richard-Greenblatt [5,6], Margaret Mokomane[2], Mosepele Mosepele[3,7], Corrado Cancedda[1,3], David M. Goldfarb[8], Ashley Styczynski [9], Gemma Parra [9], Rachel M. Smith[9], Naledi Mannathoko [2‡], Jonathan Strysko [1,10‡]

1 Botswana-University of Pennsylvania Partnership, Gaborone, Botswana, 2 Faculty of Health Sciences, School of Allied Health, University of Botswana, Gaborone, Botswana, 3 Division of Infectious Diseases, Department of Medicine, Perelman School of Medicine, University of Pennsylvania, Philadelphia, Pennsylvania, United States of America, 4 Center for Clinical Epidemiology and Biostatistics, Department of Biostatistics, Epidemiology, and Informatics, Perelman School of Medicine, University of Pennsylvania, Philadelphia, Pennsylvania, United States of America, 5 Department of Pediatric Laboratory Medicine and Pathology, The Hospital for Sick Children, Toronto, Canada, 6 Department of Laboratory Medicine and Pathobiology, University of Toronto, Toronto, Canada, 7 Department of Internal Medicine, Faculty of Medicine, University of Botswana, Gaborone, Botswana, 8 Department of Pathology and Laboratory Medicine, BC Children's Hospital, Vancouver, Canada, 9 Division of Healthcare Quality and Promotion, United States of America Centers for Disease Control and Prevention, Atlanta, Georgia, United States of America, 10 Division of General Pediatrics, Children's Hospital of Philadelphia, Perelman School of Medicine, University of Pennsylvania, Philadelphia, Pennsylvania, United States of America

◉ These authors contributed equally to this work and are considered co-first authors.
‡ These authors contributed equally to this work and are considered co-senior authors.
* zankere2016@gmail.com

## Abstract

Extended-spectrum cephalosporin-resistant Enterobacterales (ESCrE) and carbapenem-resistant Enterobacterales (CRE) are significant contributors to infection-related mortality in low- and middle-income countries. Colonization with ESCrE and/or CRE can precede infection and serve as a reservoir for transmission within healthcare facilities and the community. We conducted a 12-month period-prevalence study, screening patients for ESCrE and CRE upon admission to a referral tertiary hospital Emergency Department in Botswana. Rectal swabs were collected within 24 hours of hospital arrival. Colonization was identified using selective culture media and confirmed using automated susceptibility testing. Associations between ESCrE/CRE colonization, and clinical and demographic variables were analysed using univariate and multivariable logistic regression. Among 802 patients, 24.1% (n = 193) were colonized with ESCrE, and 1.5% (n = 12) with CRE. ESCrE colonization was associated with recent hospitalization (within

**Data availability statement:** All data are fully available without restriction.

**Funding:** This study was supported by the U.S. Centers for Disease Control and Prevention (CDC) through the Antibiotic Resistance in Communities and Hospitals (ARCH) consortium. Funding was awarded to University of Pennsylvania, with Principle Investigators EL, CC and JS under CDC cooperative agreement [Award number: U3HCK000015-01-00, Federal Award Identification Number: NU3HCK000015, Award Date: 16 Sep, 2021]. The funder's website is https://www.cdc.gov. CDC science advisors had a role in study design and preparation of the manuscript.

**Competing interests:** The authors have declared that no competing interests exist.

the last six months) (aOR 1.76, 95% CI 1.11-2.79), borehole water use (aOR 3.95, 95% CI 1.12-13.87), indwelling medical devices (aOR 2.19, 95% CI 1.08-4.48), and age < 1 year (aOR 2.09, 95% CI 1.32-3.30). CRE colonization was associated with antiretroviral drug use (cOR 6.60, 95% CI 1.72-25.36). Infants (<1 year) had over three times the odds of ESCr/CR-*Klebsiella* spp. colonization compared to adults (cOR 3.60, 95% CI 1.82-7.13). Infant age, recent healthcare exposure, indwelling medical devices, and borehole water use were key risk factors for ESCrE colonization, highlighting the need for targeted infection prevention strategies in Botswana. The identified potential association between CRE colonization and antiretroviral drug use warrants further investigation to elucidate any possible links and drivers between HIV care and antimicrobial resistance.

## Introduction

Antimicrobial resistance (AMR) is an escalating global health crisis, with low- and middle-income countries (LMICs) in sub-Saharan Africa and South Asia bearing the highest burden of AMR-related deaths worldwide [1]. The rising incidence of AMR-related illness in these regions is exacerbated by antibiotic misuse, a shrinking healthcare workforce, ongoing transmission of pathogens due to poor infection prevention and control practices, and limited diagnostic and public health capacity to detect, prevent, and treat AMR infections [2]. A deeper understanding of AMR epidemiology is needed in LMICs to mitigate this crisis [3].

Extended-spectrum cephalosporin-resistant Enterobacterales (ESCrE) and carbapenem-resistant Enterobacterales (CRE) are among the most critical bacterial threats, with limited treatment options and high mortality and morbidity, particularly in LMICs [4,5]. ESCrE has been found to be a leading cause of urinary tract and bloodstream infections in sub-Saharan Africa [6–8]. In Botswana, extended-spectrum cephalosporin-resistant *Klebsiella pneumoniae* is the leading cause of bloodstream infections among hospitalized neonates [9]. The rise of *K. pneumoniae* infections in paediatric populations extends beyond Botswana, with *K. pneumoniae* being identified as a leading cause of death among children under two years, with multidrug resistance being commonly reported [10,11]. Similarly, infections due to CRE are on the rise in sub-Saharan Africa, often implicated in healthcare facility outbreaks, mortality from which exceeds 36% [12–15].

ESCrE and CRE infections may be preceded by asymptomatic intestinal colonization, and colonized patients may serve as reservoirs for transmission [16,17]. Unlike surveillance of laboratory-confirmed infections, colonization studies can help identify both patient- and population-level risk factors specific to colonization and transmission of multidrug-resistant organisms [18]. A 2016 meta-analysis estimated the global prevalence of ESCrE colonization at 14%, with an annual increase of 5.4% [19]. However, ESCrE/CRE colonization varies regionally, with higher prevalence in LMICs [20,21]. In Botswana, community ESCrE and CRE colonization was 27% and 1.7%, respectively, in 2020 [22,23].

Risk factors for acquisition of ESCrE/CRE colonization in healthcare facilities include prolonged hospitalization, antibiotic exposure, and broad-spectrum antibiotic use [24,25]. Community risk factors vary, but prior studies in LMICs suggest that healthcare exposure increases the odds of ESCrE colonization [23,25].

We conducted a one-year period-prevalence study with a case-control analysis to identify risk factors for colonization with ESCrE and CRE colonization among patients being admitted to a referral hospital in Botswana.

## Methods

This work was part of the United States Centers for Disease Control and Prevention (CDC)-supported Antibiotic Resistance in Communities and Hospitals (ARCH) consortium evaluating colonization with clinically significant antimicrobial-resistant organisms across six countries.

### Ethics statement

Ethical approval was obtained from the Institutional Review Boards at the University of Pennsylvania (IRB# 2022–851492), the University of Botswana, the Health Research and Development Committee at Botswana's Ministry of Health (2022-HPDME18/13/1), and the study healthcare facility (2022–2/2A (7)/201). A waiver of written consent was approved by the Institutional Review Boards and verbal consent was obtained either from the patient themselves or from a parent, guardian, healthcare proxy, or member of the nursing staff prior to participation. Consent was documented and witnessed by study personnel.

### Study design

We conducted a one-year prevalence study with a case control analysis from June 15th 2022–15th June 2023 at a 530-bed tertiary referral hospital serving southern Botswana. Patients admitted through the Emergency Department to any hospital ward were enrolled, thus excluding patients directly admitted to inpatient wards, including maternity wards. Patients transferred to the Emergency Department from other healthcare facilities, such as clinics and district hospitals, were included but were categorized as having a recent healthcare exposure. Patients were excluded if they were in the Emergency Department for over 24 hours before enrolment.

### Data collection and processing

Participants underwent ESCrE and CRE screening via rectal swabbing within 24 hours of hospital arrival. Specimens were collected using flocked swabs, transported using Eswab® (COPAN), and inoculated on chromogenic media selective for ESCrE (CHROMAgar ESBL) and CRE (CHROMAgar mSuperCARBA) within 24 hours of sample collection. If same-day inoculation was not possible, samples were stored at 2–4°C and inoculated within 48 hours of sample collection. Suspected ESCrE and CRE colonies were sub-cultured on nutrient agar to ensure purity and underwent automated identification and antimicrobial sensitivity testing (AST) using the VITEK2 (Biomerieux) system and were interpreted using Clinical Laboratory Standards Institute M100 34th edition breakpoints [26]. Isolates that could not be identified or were missing AST results were removed. Isolates resistant to cefotaxime (MIC ≥ 4 µg/mL) and/or ceftazidime (MIC ≥ 16 µg/mL) were classified as ESCrE. CRE was defined as isolates resistant to any carbapenem (MIC ≥ 2 µg/ml for ertapenem, or MIC ≥ 4 µg/ml meropenem or imipenem).

### Variable definitions

Demographic data, including age, sex, household size, and location of residence at admission, were obtained through chart review and patient interview. Recent hospitalization was defined as overnight hospital admission, including facility births, within the six months preceding enrolment. Recent antibiotic exposure was defined as the use of any antibiotic

within the preceding 30 days, with the agent names recorded, if known. Indwelling devices included urinary catheters, endotracheal tubes, and central venous catheters. Participants reported their primary water source (well/borehole vs. municipal) and sanitation (pit latrines vs. flush toilet). Recent livestock exposure was defined as contact with livestock farms or animals within the 30 days preceding enrolment. Season of swab collection was categorized as rainy/warm (November-March) or dry/cool season (April-October).

## Statistical analysis

Participant data were cleaned, and incomplete records were removed from the analysis. Incomplete records were defined as missing or erroneous values that could impact the validity of the findings. Ages were grouped into five categories: infants (<1 year), young children (1–5 years), older children (6–17 years), adults (18–50 years), and older adults (>50 years). Household size was grouped into three categories (1–3, 4–6, or ≥7 individuals). Logistic regression analyses identified risk factors for ESCrE and CRE colonization in two separate models. Crude odds ratios (cOR) with 95% confidence intervals (CIs) were calculated, and the associated p-values were assessed to determine statistical significance. A multivariable logistic regression model was performed for each pathogen, yielding adjusted odds ratios (aOR) with 95% CIs. Variables were included in the multivariable model based on *a priori* hypothesis (i.e., sex) or if they had p-values ≤ 0.1 in univariate analysis.

Duplicate organisms, defined as isolates recovered from the same swab growing on both CHROMagar ESBL and mSuperCARBA with identical species and AST patterns were counted once in species-level analysis but included in overall ESCrE and CRE prevalence calculations. A secondary analysis evaluated associations between the three most common Enterobacterales species and collected variables. STATA version 17.0 was used to perform all analyses.

## Results

Among 1470 eligible patients, 842 participants were enrolled; 802 had complete data and were included in the analysis. The median participant age was nine years (range: 7 days to 91 years, interquartile range [IQR] 1.1-34 years); 52.4% (n = 420) were male.

### ESCrE prevalence and associations

The overall ESCrE colonization prevalence was 24.1% (n = 193). Age was strongly associated with ESCrE colonization (p < 0.01); compared to adults, infants had more than twice the odds, while older children had 60% reduced odds (**Table 1**). Recent hospitalization, chronic disease, indwelling devices, and well/borehole water use were significant risk factors. There were no associations with facility transfer, recent antibiotics, cephalosporins, carbapenems, or antiretroviral drugs.

### Multivariable analysis *of* ESCrE associations

After adjusting for hospitalization, sex, water source, indwelling device, and chronic disease, infant age was still strongly associated with ESCrE-colonization, while older children had 56% reduced odds of being ESCrE-colonized when compared to adults. Participants reporting recent hospitalization had a 76% increased odds of being colonized with ESCrE, and those who had an indwelling device had more than twice the odds. Those who used a well or borehole as their primary water source had almost four times the odds of being ESCrE-colonized when compared to those who used municipal water. There was no association between ESCrE colonization and known pre-existing chronic diseases in the adjusted model, nor was there an association with sex.

### CRE prevalence and associations

The prevalence of CRE colonization was 1.5% (n = 12). Table 2 shows baseline descriptive analyses for CRE colonization using crude ORs. People with exposure to antiretroviral drugs had 6.6 times the odds of having CRE colonization

**Table 1. Association of participant demographics and ESCrE colonization among patients admitted to a tertiary care hospital, Botswana, 2022-2023.**

| Variable | Category | Observations n=802 (%) | ESCrE colonized n=193 (%) | Not ESCrE colonized n=609 (%) | Crude Odds Ratio (95% CI) | P-value** | Adjusted Odds Ratio (95% CI) |
|---|---|---|---|---|---|---|---|
| Sex | Female | 382 (47.6) | 89 (23.3) | 293 (76.7) | REF | 0.63 | REF |
| | Male | 420 (52.4) | 104 (24.8) | 316 (75.2) | 1.08 (0.78-1.50) | | 1.02 (0.73-1.44) |
| Age (years) | 18-50 | 270 (33.7) | 57 (21.1) | 213 (78.9) | REF | **0.0001** | REF |
| | <1 | 158 (19.7) | 56 (35.4) | 102 (64.6) | **2.05 (1.32-3.18)** | | **2.09 (1.32-3.30)** |
| | 1-5 | 196 (24.4) | 49 (25.0) | 147 (75.0) | 1.24 (0.80-1.93) | | 1.31 (0.83-2.05) |
| | 6-17 | 82 (10.2) | 8 (9.8) | 74 (90.2) | **0.40 (0.18-0.89)** | | **0.44 (0.20-0.97)** |
| | >50 | 94 (11.7) | 22 (23.4) | 72 (76.6) | 1.14 (0.65-2.00) | | 0.97 (0.53-1.80) |
| Pre-existing chronic disease* | Not known | 769 (95.9) | 180 (23.4) | 589 (39.4) | REF | **0.05** | |
| | Known | 33 (4.1) | 13 (76.6) | 20 (60.6) | **2.13 (1.04-4.36)** | | |
| Indwelling device | No | 764 (95.3) | 178 (23.3) | 586 (76.7) | REF | **0.03** | REF |
| | Yes | 38 (4.7) | 15 (39.5) | 23 (60.5) | **2.15 (1.10-4.20)** | | **2.19 (1.08-4.48)** |
| Antibiotics (past 30 days) | No | 648 (80.8) | 158 (24.4) | 490 (75.6) | REF | 0.88 | |
| | Yes | 147 (18.3) | 35 (23.8) | 112 (76.2) | 0.97 (0.64-1.47) | | |
| | Not sure | 7 (0.9) | 0 (0.0) | 7 (100.0) | -- | | |
| Cephalosporin (past 30 days) | No | 740 (92.3) | 179 (24.2) | 561 (75.8) | REF | 0.77 | |
| | Yes | 62 (7.7) | 14 (22.6) | 48 (77.4) | 0.91 (0.49-1.70) | | |
| Carbapenem (past 30 days) | No | 800 (99.8) | 193 (24.1) | 607 (75.9) | REF | -- | |
| | Yes | 2 (0.2) | 0 (0.0) | 2 (100.0) | -- | | |
| Antiretroviral drugs (past 30 days) | No | 761 (94.9) | 181 (23.8) | 580 (76.2) | REF | 0.43 | |
| | Yes | 41 (5.1) | 12 (29.3) | 29 (70.7) | 1.32 (0.66-2.65) | | |
| Hospitalized in past 6 months | No | 693 (86.4) | 155 (22.4) | 538 (77.6) | REF | **0.004** | |
| | Yes | 103 (12.8) | 37 (35.9) | 66 (64.1) | **1.94 (1.25-3.02)** | | |
| Referred from a clinic or hospital | No | 98 (12.2) | 19 (19.4) | 79 (80.6) | REF | 0.24 | |
| | Yes | 701 (87.4) | 173 (24.7) | 528 (75.3) | 1.36 (0.80-2.31) | | |
| Household size | 1-3 | 302 (37.7) | 78 (25.8) | 224 (74.2) | REF | 0.62 | REF |
| | 4-6 | 366 (45.6) | 86 (23.5) | 280 (76.5) | 0.88 (0.62-1.26) | | **1.76 (1.11-2.79)** |
| | 7 or more | 129 (16.1) | 28 (21.7) | 101 (78.3) | 0.80 (0.49-1.30) | | |
| Water Source | Municipal water | 784 (97.8) | 186 (23.7) | 598 (76.3) | REF | **0.03** | REF |
| | Well/Borehole | 11 (1.4) | 6 (54.6) | 5 (45.4) | **3.86 (1.16-12.78)** | | **3.95 (1.12-13.87)** |
| | Other | 4 (0.5) | 0 (0.0) | 4 (100.0) | -- | | |
| Sanitation | Flush toilet | 520 (64.8) | 122 (23.5) | 398 (76.5) | REF | 0.66 | |
| | Pit latrine | 262 (32.7) | 67 (25.6) | 195 (74.4) | 1.12 (0.79-1.58) | | |
| | Other | 17 (2.1) | 3 (17.6) | 14 (82.4) | 0.70 (0.20-2.47) | | |
| Livestock exposure in 30 days | No | 657 (81.9) | 161 (24.5) | 496 (75.5) | REF | 0.82 | |
| | Yes | 131 (16.3) | 29 (22.1) | 102 (77.9) | 0.88 (0.56-1.37) | | |
| | Not Sure | 14 (1.8) | 3 (21.4) | 11 (78.6) | 0.84 (0.23-3.05) | | |
| Season of swab collection | Dry | 560 (69.8) | 127 (22.7) | 433 (77.3) | REF | 0.16 | |
| | Rainy | 242 (30.2) | 66 (27.3) | 176 (72.7) | 1.28 (0.90-1.80) | | |

*Pre-existing conditions were defined as any of the following: chronic kidney disease, chronic obstructive pulmonary disease, diabetes mellitus, coronary artery disease/congestive heart failure, hypertension, neurodevelopmental impairment, substance abuse/dependence, and malignancy.

**P-values <0.05 were considered statistically significant.

**Table 2. Association of participant demographics and exposure with CRE colonization among patients admitted to a tertiary care hospital, Botswana, 2022–2023.**

| Variable | Category | Observations n=802 (%) | CRE colonized n=12(%) | Not CRE colonized n=789 (%) | Crude odds ratio (95% CI) | P-value** |
|---|---|---|---|---|---|---|
| Sex | Female | 382 (47.6) | 6 (1.6) | 376 (98.4) | REF | 0.87 |
| | Male | 420 (52.4) | 6 (1.4) | 414 (98.6) | 0.91 (0.29-2.84) | |
| Age (years) | 18-50 | 270 (33.7) | 3 (1.1) | 267 (98.9) | REF | 0.83 |
| | <1 | 158 (19.7) | 3 (1.9) | 155 (98.1) | 1.72 (0.34-8.64) | |
| | 1-5 | 196 (24.4) | 4 (2.0) | 192 (98.0) | 1.85 (0.41-8.38) | |
| | 6-17 | 82 (10.2) | 0 (0.0) | 82 (100.0) | -- | |
| | >50 | 94 (11.7) | 2 (2.1) | 92 (97.9) | 1.93 (0.32-11.76) | |
| Pre-existing chronic disease* | Not known | 769 (95.9) | 11 (1.4) | 758 (98.6) | REF | 0.51 |
| | Known | 33 (4.1) | 1 (3.0) | 32 (97.0) | 2.15 (0.27-17.19) | |
| Indwelling device | No | 764 (95.3) | 11 (1.4) | 753 (98.6) | REF | 0.59 |
| | Yes | 38 (4.7) | 1 (2.6) | 37 (97.4) | 1.85 (0.23-14.71) | |
| Antibiotics (past 30 days) | No | 648 (80.8) | 9 (1.4) | 639 (98.6) | REF | 0.57 |
| | Yes | 147 (18.3) | 3 (2.0) | 144 (98.0) | 1.48 (0.40-5.53) | |
| | Not sure | 7 (0.9) | 0 (0.0) | 7 (100.0) | -- | |
| Cephalosporin (past 30 days) | No | 740 (92.3) | 11 (1.5) | 729 (98.5) | REF | 0.94 |
| | Yes | 62 (7.7) | 1 (1.6) | 61 (98.4) | 1.09 (0.14-8.56) | |
| Carbapenem (past 30 days) | No | 800 (99.8) | 12 (1.5) | 788 (98.5) | REF | -- |
| | Yes | 2 (0.2) | 0 (0.0) | 2 (100.0) | -- | |
| Antiretroviral drugs (past 30 days) | No | 761 (94.9) | 9 (1.2) | 752 (98.8) | REF | **0.02** |
| | Yes | 41 (5.1) | 3 (7.3) | 38 (92.7) | **6.60 (1.72-25.36)** | |
| Hospitalized in past 6 months | No | 693 (86.4) | 10 (1.4) | 683 (98.6) | REF | 0.68 |
| | Yes | 103 (12.9) | 1 (1.0) | 102 (99.0) | 0.67 (0.08-5.29) | |
| Referred from a clinic or hospital | No | 98 (12.2) | 1 (1.0) | 97 (99.0) | REF | 0.66 |
| | Yes | 701 (87.4) | 11 (1.6) | 690 (98.4) | 1.55 (0.20-12.11) | |
| Household size | 1-3 | 302 (37.7) | 3 (1.0) | 299 (99.0) | REF | 0.58 |
| | 4-6 | 366 (45.6) | 5 (1.4) | 361 (98.6) | 1.38 (0.33-5.82) | |
| | 7 or more | 129 (16.1) | 3 (2.3) | 126 (97.7) | 2.37 (0.47-11.92) | |
| Water Source | Municipal water | 784 (97.8) | 11 (1.4) | 773 (98.6) | REF | -- |
| | Well/Borehole | 11 (1.4) | 0 (0.0) | 11 (100.0) | -- | |
| | Other | 4 (0.5) | 0 (0.0) | 4 (100.0) | -- | |
| Sanitation | Flush toilet | 520 (64.8) | 9 (1.7) | 511 (98.3) | REF | 0.25 |
| | Pit latrine | 262 (32.7) | 2 (0.8) | 260 (99.2) | 0.44 (0.09-2.04) | |
| | Other | 17 (2.1) | 0 (0.0) | 17 (100.0) | -- | |
| Livestock exposure in 30 days | No | 657 (81.9) | 11 (1.7) | 646 (98.3) | REF | 0.24 |
| | Yes | 131 (16.3) | 0 (0.0) | 131 (100.0) | -- | |
| | Not Sure | 14 (1.75) | 1 (7.1) | 13 (92.9) | 4.52 (0.54-37.62) | |
| Season of swab collection | Dry | 560 (69.8) | 8 (1.4) | 552 (98.6) | REF | 0.81 |
| | Rainy | 242 (30.2) | 4 (1.6) | 238 (98.4) | 1.16 (0.34-3.89) | |

*Pre-existing conditions were defined as any of the following: chronic kidney disease, chronic obstructive pulmonary disease, diabetes mellitus, coronary artery disease/congestive heart failure, hypertension, neurodevelopmental impairment, substance abuse/dependence, and malignancy.

**P-values <0.05 were considered statistically significant

(p = 0.02) compared to those not on antiretroviral drugs. Given the low number of CRE-colonized patients, a multivariable analysis was not performed.

## Stratified analysis *by* bacterial species

Of 237 isolates confirmed as either ESCrE or CRE, the majority were *E. coli* (124; 52.3%) and *Klebsiella* spp. (69; 29.1%). Others included: *Enterobacter* spp. (13; 7.2%), *Serratia* spp. (14; 5.9%), *Raoultella* spp. (7; 3.0%), *Citrobacter* spp. (5; 2.1%), and *Kluyvera cryocrescens* (1; 0.4%) (Fig 1). In a crude logistic analysis of the three major species groups identified (*E. coli*, *Klebsiella* spp, *Enterobacter* spp), participants whose swab collection occurred in the rainy season had 49% increased odds of being colonized with *E. coli* when compared to those whose swab collection occurred in the dry season (OR 1.49, 95%CI 1.0-2.23, p = 0.05) (Table 3). Infants had more than three times the odds of being colonized with *Klebsiella* spp. when compared to adults (OR 3.60, 95%CI 1.82-7.13, p < 0.001) (Table 4). Participants who reported that they were hospitalised within the previous 6 months had more than three times the odds of being colonized with *Klebsiella* spp. (OR 3.28, 95%CI 1.81-5.96, p < 0.001), and those whose water source was a well/borehole had more than seven times the odds of being colonized with *Klebsiella* spp. when compared to those whose water source was municipal (OR 7.43, 95%CI 2.11-26.14, p = 0.006). No associations were observed with *Enterobacter* spp. colonization (S1 Table).

## Discussion

In this study conducted at a tertiary hospital in Botswana, we found that 24.1% and 1.5% of patients were colonized with ESCrE and CRE, respectively, at the time of admission, indicating a substantial proportion of patients already harbouring multidrug-resistant organisms upon arrival to the hospital. In addition to recent healthcare exposure and indwelling medical devices, infant age emerged as a major independent risk factor for ESCrE colonization. We also found that the use of untreated borehole water was associated with an increased odds of ESCrE colonization, further emphasizing community-level environmental drivers of AMR.

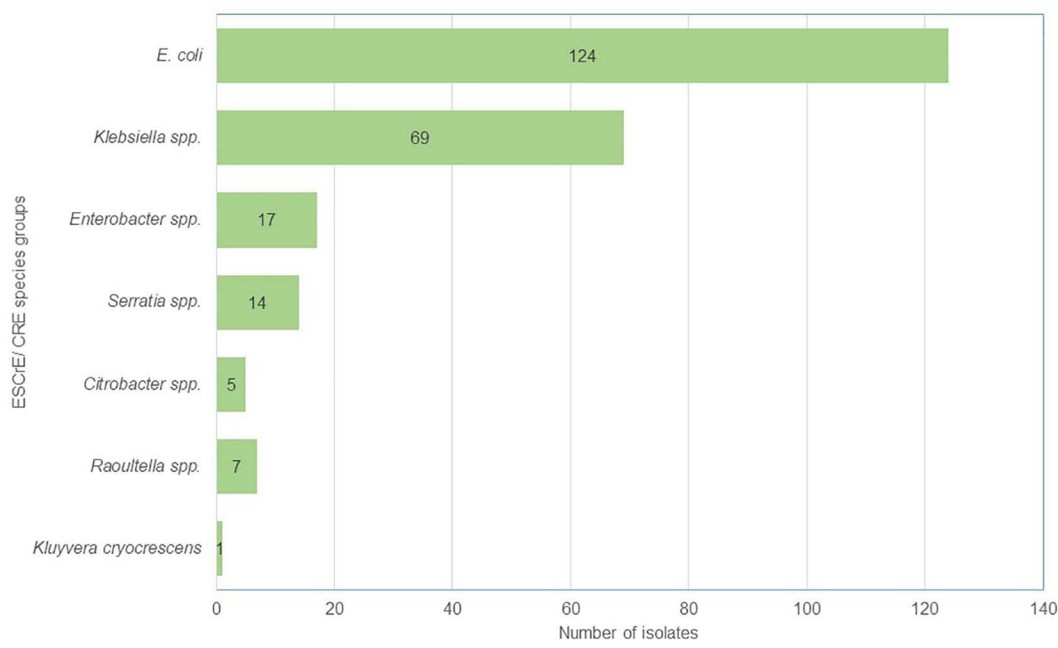

**Fig 1. Stratified analysis by bacterial species.**

**Table 3. Association of participant demographics and exposure with ESCrE/CRE *E. Coli* colonization among patients admitted to a tertiary care hospital, Botswana, 2022-2023.**

| Variable | Category | Observations n = 802 (%) | *E. Coli* n = 122 (%) | Non-*E. Coli* n = 680 (%) | Odds ratio (95% CI) | P-value** |
|---|---|---|---|---|---|---|
| Sex | Female | 382 (47.6) | 58 (15.2) | 324 (84.8) | REF | 0.98 |
| | Male | 420 (52.4) | 64 (15.4) | 356 (84.8) | 1.00 (0.68-1.48) | |
| Age (years) | 18-50 | 270 (33.7) | 44 (16.3) | 226 (83.7) | REF | 0.11 |
| | <1 | 158 (19.7) | 25 (15.8) | 133 (84.2) | 0.96 (0.56-1.65) | |
| | 1-5 | 196 (24.4) | 30 (15.3) | 166 (84.7) | 0.93 (0.56-1.54) | |
| | 6-17 | 82 (10.2) | 5 (6.1) | 77 (93.9) | 0.33 (0.13-0.87) | |
| | >50 | 94 (11.7) | 17 (18.1) | 77 (81.9) | 1.13 (0.61-2.10) | |
| Pre-existing chronic disease* | Not known | 769 (95.9) | 114 (14.8) | 655 (85.2) | REF | 0.17 |
| | Known | 33 (4.1) | 8 (24.2) | 25 (75.8) | 1.84 (0.81-4.18) | |
| Indwelling device | No | 764 (95.3) | 113 (14.8) | 651 (85.2) | REF | 0.16 |
| | Yes | 38 (4.7) | 9 (23.7) | 29 (76.3) | 1.79 (0.82-3.88) | |
| Antibiotics (past 30 days) | No | 648 (80.8) | 99 (15.3) | 549 (84.7) | REF | 0.91 |
| | Yes | 147 (18.33) | 23 (15.6) | 124 (84.4) | 1.03 (0.63-1.68) | |
| | Not sure | 7 (0.87) | 0 (0.0) | 7 (100.0) | -- | |
| Cephalosporin (past 30 days) | No | 740 (92.3) | 112 (15.1) | 628 (84.9) | REF | 0.84 |
| | Yes | 62 (7.7) | 10 (16.1) | 52 (83.9) | 1.08 (0.53-2.18) | |
| Carbapenem (past 30 days) | No | 800 (99.8) | 122 (155.2) | 678 (84.8) | REF | -- |
| | Yes | 2 (0.2) | 0 (0.0) | 2 (100.0) | -- | |
| Antiretroviral drugs (past 30 days) | No | 761 (94.9) | 115 (15.1) | 646 (84.9) | REF | 0.74 |
| | Yes | 41 (5.1) | 7 (17.1) | 34 (82.9) | 1.16 (0.50-2.67) | |
| Hospitalized in past 6 months | No | 693 (86.4) | 107 (15.4) | 586 (84.6) | REF | 0.82 |
| | Yes | 103 (12.8) | 15 (14.6) | 88 (85.4) | 0.93 (0.52-1.68) | |
| Referred from a clinic or hospital | No | 98 (12.2) | 10 (10.2) | 88 (89.8) | REF | 0.12 |
| | Yes | 701 (87.4) | 112 (16.0) | 589 (84.0) | 1.67 (0.84-3.32) | |
| Household size | 1-3 | 302 (37.7) | 57 (18.9) | 245 (81.1) | REF | 0.08 |
| | 4-6 | 366 (45.6) | 50 (13.7) | 316 (86.3) | 0.68 (0.45-1.03) | |
| | 7 or more | 129 (16.1) | 15 (11.6) | 114 (88.4) | 0.57 (0.31-1.04) | |
| Water Source | Municipal water | 784 (97.8) | 118 (15.0) | 666 (85.0) | REF | 0.08 |
| | Well/Borehole | 11 (1.4) | 4 (36.4) | 7 (63.6) | 3.22 (0.93-11.19) | |
| | Other | 4 (0.5) | 0 (0.0) | 4 (100.0) | -- | |
| Sanitation | Flush toilet | 520 (64.8) | 80 (15.4) | 440 (84.6) | REF | 0.48 |
| | Pit latrine | 262 (32.7) | 41 (15.6) | 221 (84.4) | 1.02 (0.68-1.54) | |
| | Other | 17 (2.1) | 1 (5.9) | 16 (94.1) | 0.34 (0.04-2.63) | |
| Livestock exposure in 30 days | No | 657 (81.9) | 99 (15.1) | 558 (84.9) | REF | 0.57 |
| | Yes | 131 (16.3) | 22 (16.8) | 109 (83.2) | 1.14 (0.69-1.88) | |
| | Not Sure | 14 (1.8) | 1 (7.1) | 13 (92.9) | 0.43 (0.06-3.35) | |
| Season of swab collection | Dry | 560 (69.8) | 76 (13.6) | 484 (86.4) | REF | **0.05** |
| | Rainy | 242 (30.2) | 46 (19.0) | 196 (81.0) | **1.49 (1.0-2.23)** | |

*Pre-existing conditions were defined as any of the following: chronic kidney disease, chronic obstructive pulmonary disease, diabetes mellitus, coronary artery disease/congestive heart failure, hypertension, neurodevelopmental impairment, substance abuse/dependence, and malignancy.

**P-values <0.05 were considered statistically significant.

**Table 4. Association of participant demographics and exposure with ESCrE/CRE *Klebsiella* spp. colonization among patients admitted to a tertiary care hospital, Botswana, 2022-2023.**

| Variable | Category | Observations n = 802 (%) | *Klebsiella* spp. n = 60 (%) | Non-*Klebsiella* n = 742 (%) | Crude Odds Ratio (95% CI) | P-value** |
|---|---|---|---|---|---|---|
| Sex | Female | 382 (47.6) | 26 (6.8) | 356 (93.2) | REF | 0.49 |
| | Male | 420 (52.4) | 34 (8.1) | 386 (91.9) | 1.21 (0.71-2.05) | |
| Age (years) | 18-50 | 270 (33.7) | 14 (5.2) | 256 (94.8) | REF | **<0.001** |
| | <1 | 158 (19.7) | 26 (16.5) | 132 (83.5) | **3.60 (1.82-7.13)** | |
| | 1-5 | 196 (24.4) | 12 (6.1) | 184 (93.9) | 1.19 (0.54-2.64) | |
| | 6-17 | 82 (10.2) | 3 (3.7) | 79 (96.3) | 0.69 (0.19-2.48) | |
| | >50 | 94 (11.7) | 4 (4.3) | 90 (95.7) | 0.81 (0.26-2.53) | |
| Pre-existing chronic disease* | Not known | 769 (95.9) | 55 (7.2) | 714 (92.8) | REF | 0.13 |
| | Known | 33 (4.1) | 5 (15.2) | 28 (84.8) | 2.32 (0.86-6.24) | |
| Indwelling device | No | 764 (95.3) | 55 (7.2) | 709 (92.8) | REF | 0.21 |
| | Yes | 38 (4.7) | 5 (13.2) | 33 (86.8) | 1.95 (0.73-5.20) | |
| Antibiotics (past 30 days) | No | 648 (80.8) | 49 (7.6) | 599 (92.4) | REF | 0.97 |
| | Yes | 147 (18.33) | 11 (7.5) | 136 (92.5) | 0.99 (0.50-1.95) | |
| | Not sure | 7 (0.87) | 0 (0.0) | 7 (100.0) | -- | |
| Cephalosporin (past 30 days) | No | 740 (92.3) | 56 (7.6) | 684 (92.4) | REF | 0.74 |
| | Yes | 62 (7.7) | 4 (6.4) | 58 (93.6) | 0.84 (0.30-2.40) | |
| Carbapenem (past 30 days) | No | 800 (99.8) | 60 (7.5) | 740 (92.5) | REF | -- |
| | Yes | 2 (0.2) | 0 (0.0) | 2 (100.0) | -- | |
| Antiretroviral drugs (past 30 days) | No | 761 (94.9) | 55 (7.2) | 706 (92.8) | REF | 0.27 |
| | Yes | 41 (5.1) | 5 (12.2) | 36 (87.8) | 1.78 (0.67-4.72) | |
| Hospitalized in past 6 months | No | 693 (86.4) | 42 (6.1) | 651 (93.9) | REF | **<0.001** |
| | Yes | 103 (12.8) | 18 (17.5) | 85 (82.5) | **3.28 (1.81-5.96)** | |
| Referred from a clinic or hospital | No | 98 (12.2) | 9 (9.2) | 89 (90.8) | REF | 0.48 |
| | Yes | 701 (87.4) | 50 (7.1) | 651 (92.9) | 0.76 (0.36-1.60) | |
| Household size | 1-3 | 302 (37.7) | 21 (7.0) | 281 (93.0) | REF | 0.70 |
| | 4-6 | 366 (45.6) | 27 (7.4) | 339 (92.6) | 1.06 (0.59-1.93) | |
| | 7 or more | 129 (16.1) | 12 (9.3) | 117 (90.7) | 1.37 (0.65-2.88) | |
| Water Source | Municipal water | 784 (97.8) | 56 (7.1) | 728 (92.9) | REF | **0.006** |
| | Well/Borehole | 11 (1.4) | 4 (36.4) | 7 (63.6) | **7.43 (2.11-26.14)** | |
| | Other | 4 (0.5) | 0 (0.0) | 4 (100.0) | -- | |
| Sanitation | Flush toilet | 520 (64.8) | 34 (6.5) | 486 (93.5) | REF | 0.35 |
| | Pit latrine | 262 (32.7) | 24 (9.2) | 238 (90.8) | 1.44 (0.84-2.49) | |
| | Other | 17 (2.1) | 2 (11.8) | 15 (88.2) | 1.90 (0.42-8.68) | |
| Livestock exposure in 30 days | No | 657 (81.9) | 50 (7.6) | 607 (92.4) | REF | 0.96 |
| | Yes | 131 (16.3) | 9 (6.9) | 122 (93.1) | 0.90 (0.43-1.87) | |
| | Not Sure | 14 (1.8) | 1 (7.1) | 13 (92.9) | 0.93 (0.12-7.28) | |
| Season of swab collection | Dry | 560 (69.8) | 41 (7.3) | 519 (92.7) | REF | 0.79 |
| | Rainy | 242 (30.2) | 19 (7.8) | 223 (92.2) | 1.08 (0.61-1.90) | |

*Pre-existing conditions were defined as any of the following: chronic kidney disease, chronic obstructive pulmonary disease, diabetes mellitus, coronary artery disease/congestive heart failure, hypertension, neurodevelopmental impairment, substance abuse/dependence, and malignancy.

**P-values <0.05 were considered statistically significant.

Infants had more than twice the odds of being ESCrE colonized and more than three times the odds of colonization with multidrug-resistant *Klebsiella* spp. as compared to adults. These data are consistent with a systematic review of ESCrE colonization in sub-Saharan Africa, which showed that infants, notably neonates, were more likely to be colonized by *K. pneumoniae* [21,27]. Although the underlying reasons for this age-related difference warrant further study, early-life healthcare exposure is likely a key factor [28]. Botswana's high rate of facility-based deliveries, at 99%, is a testament to successful maternal and newborn health efforts, which have significantly improved survival [29]. However, it is possible that ESCrE colonization, particularly with *K. pneumoniae,* occurs in the permissive environment of the neonatal gut shortly after hospital birth and persists beyond six months, even in those who avoid subsequent hospitalization [30,31]. It is also possible that frequent non-inpatient healthcare exposures, such as clinic visits for weight checks and immunizations (which were not captured as variables in this study) may impact the risk of colonization in infants [32]. Understanding colonization dynamics related to healthcare exposures in the neonatal period and later infancy is crucial to inform institutional infection prevention strategies.

In addition to increased healthcare exposures experienced by infants, high colonization prevalence may be due to the possibility that *Klebsiella* spp. survive with minimal fitness cost in the human infant gut as compared with adults [31]. Infant feeding practices (i.e., breastfeeding and choice of supplementary foods) were not systematically surveyed as part of this study and early use of infant formula has been shown to be associated with higher burden of colonization with *K. pneumoniae* AMR genes [33,34]. Considering the importance of *K. pneumoniae* as a major contributor to mortality in children under two years, strategies to prevent infant acquisition in healthcare settings and reduce abundance of *Klebsiella* spp. colonization, particularly drug-resistant strains, in this age group are warranted. One randomized controlled trial demonstrated that adding synbiotics to formula-fed infants over prebiotics alone reduced *Klebsiella* spp. abundance in gut microbiota [35]. It is possible that human milk feeding may act in the same way, but more research is needed to elucidate this effect [34].

Healthcare exposures, including recent hospitalization and the presence of indwelling devices, were associated with an increased odds ESCrE colonization in this analysis. This finding echoes the results of a study in Kenya, which demonstrated that increased healthcare contact significantly raised the likelihood of colonization, with the probability rising by about 30% from the lowest to highest levels of contact [27]. Moreover, the presence of indwelling devices more than doubled the odds of ESCrE colonization in this analysis, highlighting the substantial risks posed by invasive procedures and devices which are prone to contamination [32]. These results reinforce the importance of stringent infection prevention and control measures, particularly for high-risk populations with frequent healthcare interactions.

The prevalence of ESCrE/CRE colonization ascertained in this study was similar to estimates reported by a previous study conducted among community participants in Botswana in 2020 [22]. This suggests that, while community ESCrE/CRE prevalence may have increased since 2020, hospital admission screening likely does not overestimate community ESCrE/CRE prevalence and may serve as a proxy measure in the future if community surveys are not feasible. Low CRE prevalence presents an opportunity for timely infection prevention and control interventions to contain the transmission of CRE in healthcare facilities. For example, in this setting, where universal screening of all inpatients may not be feasible, patients identified at high risk of CRE colonization or infection could be selectively screened for CRE, and inpatient isolation precautions could be implemented to help prevent transmission [15].

Although low CRE colonization resulted in inadequate statistical power to detect associations, the crude association between CRE colonization and antiretroviral drug exposure warrants further exploration. However, because HIV status and viral load were not captured in this study, it remains unclear whether the observed effect is attributable to the use of antiretrovirals (for either therapy or prophylaxis), more frequent outpatient healthcare contact, the underlying immunological effects of HIV, or any combination of these factors. HIV-related immune dysregulation is known to influence gut microbiota composition and susceptibility to colonization, [36] while specific antiretroviral drugs could independently impact microbial dynamics [37–39]. This finding highlights the need for additional research to disentangle the roles of HIV infection and antiretroviral drugs in driving CRE colonization.

We observed in this study that participants whose water source was a well/borehole had more than three times the odds of being colonized with an ESCrE, when compared to those with a municipal water source. This may reflect the fact that well/borehole water is typically untreated, while municipal water is treated, and groundwater may be exposed to sewage contamination [40–42]. Untreated well/borehole water may become contaminated if stored for long periods of time, but water storage methods were not explored in this study. Alternatively, water source may be a surrogate for wealth index or residence in urban/rural setting, which our study did not specifically capture.

We detected no significant association between recent antibiotic use and colonization with ESCrE, which is counterintuitive as the selective pressure of antibiotics is known to induce antimicrobial resistance. Although this analysis may be affected by impaired patient recall of prior antibiotic use, it is worth noting that other studies in sub-Saharan Africa, notably a point prevalence survey in Kenya enrolling 1416 community participants, also failed to detect an association between ESCrE colonization and recent antibiotic use [32]. This may be reflective of increased circulation of drug-resistant strains in the community, potentially driven by indirect antibiotic exposure through food, water, or herbal medicines. Some studies have used antibiotic residue detected in urine to demonstrate that indirect antibiotic exposures may be a potentially insidious driver of AMR [43].

A strength of this study is that it ran for an entire year and thus was poised to detect seasonal variations, if such existed. We did not identify any association with seasonality for either ESCrE or CRE colonization, except for extended-spectrum cephalosporin-resistant *E. coli*. This is in contrast to a cohort study in Malawi, which found that ESCrE colonization, particularly with *K. pneumoniae,* was higher in the wet season, possibly driven by animal–human–waste interactions being more common in the wet season [44]. Seasonal variation may be regionally-specific and may reflect the nuance of local ecology, climate, and water, sanitation, and hygiene infrastructure. For example, in previous studies in Botswana, community participants were more likely to be ESCrE-colonized if they tended to livestock, an association which we did not observe in this study [22,23].

## Limitations

There were several potential limitations to our study. First, the study was conducted at a single tertiary referral hospital in Botswana and enrolled patients seeking healthcare, which may not represent the broader population or healthcare settings in other regions of the country or sub-Saharan Africa. Children (<18 years) comprised the majority of participants, likely due to our exclusion of patients with ED stays of >24 hours; paediatric patients are typically admitted more rapidly than adults, potentially introducing selection bias. Second, while we controlled for numerous variables, there may be unmeasured confounders such as non-inpatient healthcare exposures (e.g., clinic visits) and indirect antibiotic exposure through food and water sources that were not captured. Additionally, the reliance on patient recall for antibiotic use could have potentially affected the accuracy of this variable. Our study may have also had limited statistical power, particularly in the CRE analysis, which was based on low numbers (n = 12).

## Conclusions

Our study identified a significant prevalence of ESCrE colonization among patients admitted to a referral hospital in Botswana, with recent hospitalization, medical devices, well/borehole water use, and infant age emerging as major risk factors. The finding that infants are at increased risk of colonization, particularly with *Klebsiella* spp., underscores the need for targeted interventions to prevent early acquisition and potential transmission of ESCrE in this vulnerable population. The low prevalence of CRE colonization found in this study represents a crucial window of opportunity to mitigate the spread of CRE by strengthening infection prevention and control efforts. Further research is needed to explore the impact of non-inpatient healthcare exposures and indirect antibiotic exposure on ESCrE/CRE colonization, as well as strategies to reduce colonization rates among high-risk populations, especially infants.

## Supporting information

**S1 Table. Association of participant demographics and exposure with ESCrE/CRE Enterobacter spp. colonization among patients admitted to a tertiary care hospital, Botswana, 2022–2023.**
(DOCX)

**S1 Data. Minimal data set.**
(XLSX)

## Author contributions

**Conceptualization:** Tshiamo Zankere, Kwana Lechiile, Ebbing Lautenbach, Rachel M. Smith, Naledi Mannathoko, Jonathan Strysko.

**Data curation:** Tshiamo Zankere, Kwana Lechiile, Kitso Mokgwathi, Nametso Tlhako, Tlhalefo D. Ntereke, Teresia Gatonye, Ebbing Lautenbach, Naledi Mannathoko, Jonathan Strysko.

**Formal analysis:** Tshiamo Zankere, Kwana Lechiile, Melissa Richard-Greenblatt, Jonathan Strysko.

**Funding acquisition:** Melissa Richard-Greenblatt, Corrado Cancedda, Ashley Styczynski, Gemma Parra, Rachel M. Smith, Jonathan Strysko.

**Investigation:** Tshiamo Zankere, Kitso Mokgwathi, Nametso Tlhako, Banno Moorad, Tlhalefo D. Ntereke, Teresia Gatonye, Naledi Mannathoko, Jonathan Strysko.

**Methodology:** Tshiamo Zankere, Kitso Mokgwathi, Nametso Tlhako, Banno Moorad, Tlhalefo D. Ntereke, Teresia Gatonye, Naledi Mannathoko, Jonathan Strysko.

**Project administration:** Corrado Cancedda, Naledi Mannathoko, Jonathan Strysko.

**Resources:** Ebbing Lautenbach, Melissa Richard-Greenblatt, Margaret Mokomane, Corrado Cancedda, Ashley Styczynski, Gemma Parra, Rachel M. Smith, Jonathan Strysko.

**Software:** Kwana Lechiile, Jonathan Strysko.

**Supervision:** Kwana Lechiile, Ebbing Lautenbach, Melissa Richard-Greenblatt, Margaret Mokomane, Mosepele Mosepele, David M. Goldfarb, Naledi Mannathoko, Jonathan Strysko.

**Validation:** Tshiamo Zankere, Kwana Lechiile, Ebbing Lautenbach, Margaret Mokomane, Ashley Styczynski, Naledi Mannathoko, Jonathan Strysko.

**Visualization:** Tshiamo Zankere, Kwana Lechiile, Nametso Tlhako, Ebbing Lautenbach, Naledi Mannathoko, Jonathan Strysko.

**Writing – original draft:** Tshiamo Zankere, Kwana Lechiile, Ebbing Lautenbach, Naledi Mannathoko, Jonathan Strysko.

**Writing – review & editing:** Tshiamo Zankere, Kwana Lechiile, Kitso Mokgwathi, Nametso Tlhako, Banno Moorad, Tlhalefo D. Ntereke, Teresia Gatonye, Ebbing Lautenbach, Melissa Richard-Greenblatt, Margaret Mokomane, Mosepele Mosepele, Corrado Cancedda, David M. Goldfarb, Ashley Styczynski, Gemma Parra, Rachel M. Smith, Naledi Mannathoko, Jonathan Strysko.

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
