## [Decision Letter · Decision Letter 0]

17 Jul 2025

PGPH-D-25-01363

Admission screening for extended-spectrum cephalosporin-resistant and carbapenem-resistant Enterobacterales colonization at a referral hospital in Botswana: A one-year period-prevalence survey, 2022-2023

Dear Dr. Tshiamo Zankere,

Thank you for submitting your manuscript to PLOS Global Public Health. After careful consideration, we feel that it has merit but does not fully meet PLOS Global Public Health’s publication criteria as it currently stands. Therefore, we invite you to submit a revised version of the manuscript that addresses the points raised during the review process.

We look forward to receiving your revised manuscript.

Kind regards,

Muhammad Asaduzzaman, MD MPH MPhil

Academic Editor

Journal Requirements:

1. In the ethics statement in the Methods, you have specified that verbal consent was obtained. Please provide additional details regarding how this consent was documented and witnessed, and state whether this was approved by the IRB.

Additional Editor Comments (if provided):

Reviewers' comments:

Reviewer's Responses to Questions

**Comments to the Author**

1. Does this manuscript meet PLOS Global Public Health’s publication criteria?

Reviewer #1: Yes

Reviewer #2: Yes

2. Has the statistical analysis been performed appropriately and rigorously?

Reviewer #1: Yes

Reviewer #2: Yes

3. Have the authors made all data underlying the findings in their manuscript fully available (please refer to the Data Availability Statement at the start of the manuscript PDF file)?

Reviewer #1: Yes

Reviewer #2: Yes

4. Is the manuscript presented in an intelligible fashion and written in standard English?

Reviewer #1: Yes

Reviewer #2: Yes

Reviewer #1: Zankere et al present results of a one year period-prevalence study of ESCRE and CRE and potential drivers among patients admittted to a hosptial in Botswana. The paper is well-written, the findings are carefully explored, and the data underpinning the conclusions are clearly presented. I have only two minor comments about the methods:

1) Lines 118-119: Why were patients excluded if they were in the ER for over 24 hours? Also, why might patients be in the ER for this long? Specifically, are there specific populaitons (e.g. teenagers) who might be kept in the ER for longer? I am wondering if this exclusion criteria might bias your final study population in some way.

2) Lines 130-133: Please include citations for these MIC cut-offs to define resistance. The MIC cut-off for ceftazidime in particular seems quite high. Is it possible you missed ESCrE with lower levels of resistance to cefotaxime or ceftazidime?

Reviewer #2: Concerns

1. Generalisability

- Single-Centre Limitation: The study is conducted at a single tertiary hospital, which may limit generalizability to other healthcare settings or the broader community in Botswana and sub-Saharan Africa.

- Selection Bias: Patients admitted through the Emergency Department may not represent all hospital admissions, especially as direct ward admissions (e.g., maternity) are excluded.

2. Exposure and Confounding Variables

- Incomplete Capture of Healthcare Exposure: The study does not account for non-inpatient healthcare exposures (e.g., outpatient or clinic visits), which may be significant, especially for infants.

- Antibiotic Exposure Assessment: Reliance on patient recall for antibiotic use could introduce misclassification bias. Consider discussing potential impact or alternative approaches (e.g., pharmacy records, biomarker testing).

3. Statistical Power for CRE Analysis

- Low CRE Prevalence: With only 12 CRE-colonised patients, the power to detect associations is limited, and multivariable analysis was not performed. This limitation is acknowledged but should be emphasised further in the discussion.

4. Interpretation of Antiretroviral Association

- Causality and Mechanism: The observed association between antiretroviral drug use and CRE colonisation is intriguing but remains unexplained—the lack of HIV status and viral load data limits interpretation. The discussion should more clearly state that causality cannot be inferred and that this finding is hypothesis-generating.

Minor Concerns and Suggestions

- Clarity in Tables: Some tables are dense; consider summarising key findings in the text or using graphical abstracts for clarity.

- Terminology: Ensure consistent use of terms (e.g., "well/borehole" vs. "well water") throughout the manuscript.

- Supplementary Data: Ensure that all supplementary tables and figures are referenced in the main text.

- Formatting: Minor typographical errors (e.g., misplaced line breaks, HTML entities like < and >) should be corrected before publication.

**Do you want your identity to be public for this peer review?** For information about this choice, including consent withdrawal, please see our Privacy Policy

Reviewer #1: No

Reviewer #2: **Yes: ** Benjamin Djoudalbaye

---

## [Decision Letter · Decision Letter 1]

9 Oct 2025

Admission screening for extended-spectrum cephalosporin-resistant and carbapenem-resistant Enterobacterales colonization at a referral hospital in Botswana: A one-year period-prevalence survey, 2022-2023

PGPH-D-25-01363R1

We are pleased to inform you that your manuscript 'Admission screening for extended-spectrum cephalosporin-resistant and carbapenem-resistant Enterobacterales colonization at a referral hospital in Botswana: A one-year period-prevalence survey, 2022-2023' has been provisionally accepted for publication in PLOS Global Public Health.

Best regards,

Muhammad Asaduzzaman, MD MPH MPhil

Academic Editor

Reviewer Comments (if any, and for reference):

Reviewer's Responses to Questions

**Comments to the Author**

Reviewer #1: All comments have been addressed

Reviewer #2: All comments have been addressed

publication criteria?

Reviewer #1: Yes

Reviewer #2: Yes

3. Has the statistical analysis been performed appropriately and rigorously?

Reviewer #1: Yes

Reviewer #2: Yes

4. Have the authors made all data underlying the findings in their manuscript fully available (please refer to the Data Availability Statement at the start of the manuscript PDF file)?

Reviewer #1: Yes

Reviewer #2: Yes

5. Is the manuscript presented in an intelligible fashion and written in standard English?

Reviewer #1: Yes

Reviewer #2: Yes

Reviewer #1: (No Response)

Reviewer #2: Strengths: The study is original, addresses an urgent public health topic with local and global relevance, and meets high scientific reporting standards. Results will inform infection control and AMR stewardship strategies across similar settings.

Compliance: The manuscript is in compliance with PLOS formatting and ethics policies, with transparent data sharing and clear reporting.

Limitations Acknowledged: While generalizability is limited and CRE analysis underpowered, the manuscript’s candid limitations section mitigates overinterpretation. The recommendation for further work on HIV and CRE, and a planned birth cohort study, demonstrates the team’s commitment to ongoing research.

**Do you want your identity to be public for this peer review?** For information about this choice, including consent withdrawal, please see our Privacy Policy

Reviewer #1: No

Reviewer #2: **Yes: ** Djoudalbaye Benjamin
